# Body Size of Female Strepsipteran Parasites (Strepsiptera, Xenidae, *Xenos*) Depends on Several Key Factors in a Vespine Wasp (Hymenoptera, Vespidae, *Vespa*)

**DOI:** 10.3390/insects15040266

**Published:** 2024-04-12

**Authors:** Kazuyuki Kudô, Wataru Oyaizu, Rikako Kusama, Yuki Yamaguchi, Shinsaku Koji

**Affiliations:** 1Laboratory of Insect Ecology, Faculty of Education, Niigata University, Niigata 950-2181, Japan; kudok72@yahoo.co.jp (W.O.); y_yamaguchi@vos.nagaokaut.ac.jp (Y.Y.); 2Laboratory of Insect Ecology, Graduate School of Science and Technology, Niigata University, Niigata 950-2181, Japan; skoji@create.niigata-u.ac.jp

**Keywords:** parasitism, body size, hornet, *Vespa analis*, *Xenos oxyodontes*

## Abstract

**Simple Summary:**

We studied whether the female body size of strepsipteran parasites of *Xenos oxyodontes* is affected by the following four factors: season, host body size, multiparasitism, and reproductive conditions of the host wasp. The cephalothorax width of female parasites did not change throughout the seasons but was strongly affected by host body size, number of parasites per host wasp, and reproductive conditions of host wasps. Our results suggested that female parasites with larger body sizes have greater potential for reproduction.

**Abstract:**

Parasite growth in hosts depends on their hosts’ nutritional status. This study examined whether the body size of the strepsipteran parasite *Xenos oxyodontes*, which parasitizes the eusocial wasp *Vespa analis*, is affected by several key factors, including host body size. We collected *V. analis* using bait traps for three years in Niigata, Japan, and evaluated the number of male and female *X. oxyodontes* parasites throughout the seasons. A total of 185 female parasites were collected, and their cephalothorax widths were measured. The widths of female parasites did not statistically vary among seasons and were positively correlated with the head widths of female host wasps but negatively correlated with the number of parasites per host wasp. In addition, we examined whether the reproductive conditions of post-hibernation solitary queens affected the cephalothorax width of female parasites. The widths of the female parasites were greater when the queens had less-developed ovaries. These results suggested that nutrient availability by female parasites depends on the host wasp and competition with conspecific individuals.

## 1. Introduction

Host–parasite interactions represent a strong selective force that negatively affects the host. Parasitism induces not only a reduction in lifespan and reproductive success but also physiological, behavioral, and morphological changes in hosts [1]. For example, parasitism by a protozoan reduces the ovary size of bumblebee queens; consequently, colony development is slower in the early season [2]. Numerous host resistance mechanisms, such as immune defense and rapid recovery after infection, have been studied [3,4].

Strepsiptera is an insect order that exhibits obligate endoparasites with unique morphological and biological adaptations. Many species belonging to this order show remarkable sexual dimorphism, i.e., winded males and neotenic females. Males fly when searching for mating partners, but females remain inside the host individuals throughout their lives. Parasitism by Strepsiptera leads to changes in the external morphology of the host Hymenoptera, such as coloration, length of hair and some body segments, and loss of gender characteristics [5]. They also induced behavioral changes in hosts in social wasp colonies. For example, Vespidae workers parasitized by strepsipterans do not perform extranidal activities, and males and new queens do not mate [6].

The biology of strepsipterans among vespine wasps was first investigated in the 1970s. For example, Matsuura [7] observed the reproductive activities of a female strepsipteran that parasitized a worker of *Vespa mandarinia* Smith. Matsuura and Yamane [6] published a book on vespine biology that described the natural enemies of wasps. *Xenos* is a strepsipteran genus whose members parasitize *Vespa* wasps [8]. Adult female wasps engage in extranidal activities, such as visiting and consuming tree sap for sustenance and providing nourishment to their larvae. In contrast, the first-instar larvae of the *Xenos* parasite await the opportunity to attach themselves to host nests. In Japan, *Xenos* spp. parasitize six of seven *Vespa* spp. across all main islands [9,10]. Nakase and Kato [11] conducted a molecular and morphological phylogenetic analysis of *Xenos* specimens collected from hornets in East Asia, showing two distinct *Xenos* spp.: *Xenos moutoni* du Buysson and *Xenos oxyodontes* Nakase and Kato. They also demonstrated that these two species exhibit differences in their host utilization patterns: *X*. *oxyodontes* targets *V*. *analis* and *Vespa simillima* Smith primarily, whereas *X. moutoni* is associated *Vespa* spp. excluding *V*. *analis*.

Empirical studies on the strepsipteran biology of vespine wasps have been conducted since the late 1990s in Japan. Makino and Yamashita [12] collected samples using bait traps and showed the levels of parasitism, differences in emergence period among sexes, and the position of the parasite inside their host wasps. Since then, the levels of parasitism in *Vespa* spp. have been extensively investigated using traps across various regions of Japan [13,14,15,16,17]. In addition, Makino et al. [18] estimated the level of parasitism by *X*. *oxyodontes* on *V*. *analis* using individuals from nests collected in Nagoya, Japan. Recently, Kudô et al. [17] investigated factors that affect the reproductive conditions of post-hibernation queens in *V*. *analis*. They showed that ovary development in queens was not associated with parasitism by *X*. *oxyodontes*.

A previous study has shown a relationship between the body size of female hosts (*V*. *analis*) and female strepsipterans (*X*. *oxyodontes*) [12]. Its results were consistent among the three vespine species in Japan and showed positive correlations between them. It further demonstrated a significant negative effect of multiparasitism (more than two parasite individuals per host) on the body size of female parasites. These results indicate the possible factors that affect parasite size. Since that study, Kudô et al. [19,20] also showed in two swarm-founding wasps that the body size of *Xenos* sp. females is affected by the body size of their hosts. However, the other two factors may also be linked to the body size of *Xenos* females that parasitize vespine wasps. The first is seasonality, that is, the emergence of *Xenos* females in host wasps. Significant variation in hosts’ body size by season was found in vespine wasps; spring queens are generally larger than workers or males [12]. This suggests that spring parasites have the largest average size. The second factor is the reproductive condition of the host wasps. A recent study on *Xenos* biology in *Polistes dominula* Christ, an independent-founding polistine wasp, showed resource intake and energy shifts by the parasites that can cause host wasps to reduce their allocation to reproduction [21]. This study examined the relationship between the reproductive conditions of host wasps and the body size of *Xenos* parasites.

We investigated whether the female body size of *X*. *oxyodontes* is affected by the following four factors: season, host body size, multiparasitism, and reproductive conditions of the host wasp. We retested whether host body size and multiparasitism were associated with the body size of female parasites in *V*. *analis*. In addition, we examined two other factors that affect parasitic body size. We collected wasp specimens using bait traps throughout the season for three years in Niigata, Japan, and determined the body size of the parasites and host wasps and the presence of multiparasitism under a microscope. In addition, we examined the relationship between ovary development in post-hibernation queens and the body size of female *X*. *oxyodontes*. *V*. *analis* is a model species used to study the size of *Xenos* females. Information on the level of *Xenos* infection has been previously studied in this species [13,16,17], and its infection levels are greater than those of other dominant vespine species [16,17]. Moreover, the existence of parasitized queens with ovarian development in *V*. *analis* enables examining the effect on the body size of female *X*. *oxyodontes* [17].

## 2. Materials and Methods

We collected *V. analis* in two distinct locations, Sakata Park (hereafter referred to as SP) (37°49′ N, 138°52′ E) and the Niigata University Campus (hereafter referred to as CNU) (37°52′ N, 138°56′ E), situated in Niigata city, central Japan, from 2009 to 2011. An approximate distance of 8 km separated these locations. The SP primarily featured *Salix babylonica* Linnaeus var. Babylonica, and *Celtis sinensis* Nakai were the dominant tree species in the SP, whereas Japanese black pines (*Pinus thunbergii* Parl.) were strategically planted to serve as windbreaks. Adult *V*. *analis* were collected using bait traps consisting of a mixture of 100 mL of water and an equal quantity of grape juice. We used clear plastic bottles with a capacity of 2000 mL, each equipped with a small square opening measuring 3 × 3 cm in the upper section of the bottle, as bait containers.

Wasps that entered the traps were entrapped within the containers and subsequently perished. The traps were affixed to trees at a height of 1.5 m above the ground and were subjected to weekly inspections annually from May to November. The bait was replenished weekly, and the trapped wasps were preserved in a 70% ethanol solution. We deployed 30 and 22 traps for the SP and CNU locations, respectively, positioned at 40–50 m intervals, primarily along the established walking trails.

*Xenos* parasites were observed inhabiting the terga of their host wasps (Makino and Yamashita, 1998). Regardless of the host species or sex, female parasites consistently occupied the space between the fifth and sixth terga of the hosts. Males were primarily located between the fourth and fifth terga, with occasional occurrences between the third and fourth terga [12]. Additionally, almost all male parasites have already emerged (see also Makino and Yamashita [12]), leaving only empty puparia in the host wasps. Upon encountering *Xenos* parasites, their positions were recorded to determine their sex. The number of parasites within the host wasp was then determined.

The head widths (HWs) of the parasitized wasps were measured using a binocular microscope (Olympus, SZ2-ILST). Subsequently, the gastric tergites of the parasitized wasps were removed to expose and measure the cephalothorax width (CW) of female parasites (see also Makino and Yamashita [12]). We selected HWs and CWs as body sizes of host wasps and female parasites, respectively, because these characters were measured as body sizes in previous studies (HW in [12,17]; Kudô et al., 2024, CW in [17]).

The ovary conditions of the parasitized overwintered female wasps were determined under a microscope. We selected parasitized wasps collected from June to July as post-hibernation individuals. *V*. *analis* queens emerge from hibernation in early to mid-May, find colonies, and produce the first batch of workers in mid-July in Niigata. Therefore, this period was sufficient for collecting queens that exhibited extranidal activity. The following two developmental levels were recognized in the ovaries of the parasitized queens: phase I with only one or two mature oocytes and phase II with fully developed ovarioles, each bearing more than two oocytes [17]. The mating status of the parasitized queens was determined under a binocular microscope in the presence of sperm in the spermatheca. However, both queens and workers exist among parasitized overwintered female wasps of *V*. *analis* because parasitized workers often overwinter [6,14,17]. Because *V*. *analis* workers do not mate [6], we regarded them as parasitized queens when mated and parasitized overwintering female wasps were found.

The CW of female parasites was analyzed using a linear mixed-effects model with month, host HW, and number of *Xenos* parasites per host individual as fixed effects and host individual ID as a random effect. The most parsimonious model was selected using the likelihood ratio test (Type II Wald chi-square test). The linear mixed model was fitted with the restricted maximum likelihood method using the lmer function in the lme4 package of R 4.3.2 (R Core Team, 2023) [22].

## 3. Results

In total, 4368 wasps of *V*. *analis* were captured from May to November over three years. After examining their sex, we obtained 185 female parasites from June to November.

The CW of female parasites was significantly affected by the HW of female wasps (*χ*^2^ = 7.80, df = 1, *p* < 0.01) and the number of parasites per individual host (*χ*^2^ = 52.05, df = 1, *p* < 0.001), whereas the effect of the month was insignificant (*χ*^2^ = 8.25, df = 5, *p* = 0.14). The CW of female parasites increased with increasing host HW, indicating that female parasites increased in size as they parasitized larger host wasps (Table 1, Figure 1). Cephalothorax width decreased as the number of parasites per host wasp increased (Table 1 and Figure 2).

The overwintered females parasitized a total of 31. Of these, 24 individuals were mated and were, therefore, regarded as parasitized queens. The remaining parasitized females consisted mainly of workers. Ovary development in the queens also affected the parasite’s body size. The cephalothorax width of *Xenos oxyodontes* in wasp queens with fully developed ovarioles (*n* = 13) was smaller than that of parasites found in queens with partially developed ovarioles (*n* = 8) (three queens were removed from the analysis because of multiparasitism) (Welch’s *t*-test, *t* = 2.180, df = 18.91, *p* = 0.042) (Figure 3).

## 4. Discussion

We examined whether the body size of the strepsipteran parasite *X. oxyodontes* is affected by several key factors. The cephalothorax width of female parasites did not change throughout the seasons but was strongly affected by host body size, number of parasites per host wasp, and reproductive conditions of host wasps.

### 4.1. Body Size of Host Wasps and Seasonality

A correlation between the size of hosts and parasites has been reported for some social wasps. Makino and Yamashita [12] reported the same correlation in three vespine species, including *V*. *analis*. How the first-instar infective larvae of the parasites sneak into nests and invade host larvae is not yet fully understood (but see Maeta and Kifune [23]). If the first-instar larvae singly invade a host larva that becomes a larger adult wasp, the parasitic larva gains more nutrients from the host individual and grows larger. For example, spring queens are generally larger than workers or males [12]; thus, female parasites in spring queens are larger. However, this hypothesis is not supported by our current results, possibly because of the existence of overwintered workers with small body sizes [17]. Therefore, these combined results suggest that the levels of nutrients gained by each parasite from host larvae did not depend on the production schedule of host colony castes but rather simply depended on the availability of nutrients in each host larva.

Size variation among the parasitized female wasps was observed in this study. It is likely that these females belong to different castes (i.e., queens and workers), although we did not identify them in this study. In social insects, body size variation within or among colonies is generally beneficial for extra- and intranidal activities, particularly in social bees (e.g., Chole et al. [24]). However, these general benefits do not apply to parasitized female wasps because *V*. *analis* parasitized by *X. oxyodontes* does not show any extranidal activities [6], and colony initiation by parasitized queens of this host species has not yet been reported [17].

### 4.2. Number of Female Parasites per Host Wasp

The number of strepsipteran parasites per host has been reported for different populations and seasons of vespine wasps in Japan [12,15,16,17]. It is a general trend that multiple parasitism by *X. oxyodontes* is not common, but its frequencies vary among populations; no multiple parasitism was observed in the Tokamachi City population [15], whereas approximately 22% of individuals were parasitized in other populations in mainland Japan [12,16]. Maeta [25] described that the number of parasite larvae of *Pseudoxenos iwatai* Esaki, a strepsipteran parasite of a solitary eumenid wasp, *Eumenes decorates* Smith, was one of the determining factors for the body size of the parasite. Makino and Yamashita [12] showed for the first time in *Xenos* spp. parasitizing vespine wasps that body size in multiple parasitism by *X. oxyodontes* was statistically smaller than that in single parasitism. Our results supported those of previous studies showing that the body size of female parasites was negatively correlated with the number of female parasites per host wasp. Kudô et al. [19,20] also showed the same relationships in two swarm-founding paper wasp species and interpreted that competition over nutrients among multiparasitizing strepsipteran larvae may occur in each host wasp.

### 4.3. Reproductive Conditions of Host Wasps

We detected a negative relationship between ovary development in host wasps and the size of female strepsipterans. This suggests that nutrients gained by parasites are used for greater ovary development in host wasps, and consequently, the size of the female parasites may be reduced. Future studies should examine whether a trade-off exists between the host and parasite relationships.

## 5. Conclusions

We showed that several factors affected the body size of *X. oxyodontes*. However, there is no information on how the body size of female parasites influences population dynamics. A single *Xenos* female generally produces many first-instar larvae. For example, Matsuura [7] counted the number of first-instar larvae produced by a single female *X*. *moutoni* that parasitized *V*. *mandarinia*. His records include 36,739 individuals. No information has been reported on the reproductive output of a single female of *X. oxyodontes*, but the number of first-instar larvae produced by a single female may be high and correlated with body size. Maeta [26] found in four species of Strepsiptera that the total number of eggs produced by a single female correlates negatively with the number of females per host. His study suggested that the number of eggs was reduced by smaller parasitic females because of multiple parasitism. To understand the population dynamics of *Xenos* parasites, future studies should investigate the reproduction of *X. oxyodontes* females.

## Figures and Tables

**Figure 1 insects-15-00266-f001:**
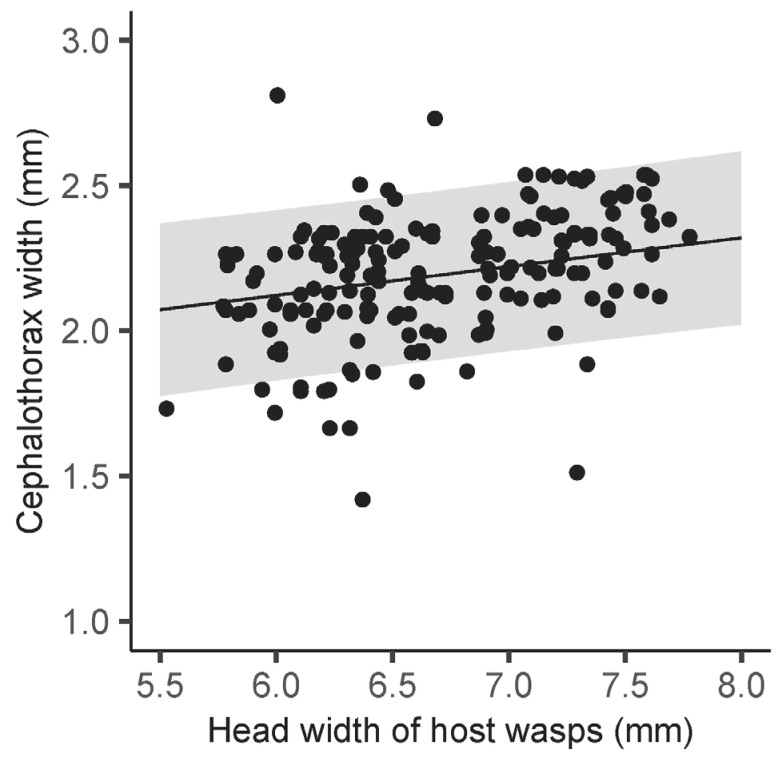
Relationship between head width of female host wasps and cephalothorax width of female parasites. The line with 95% confidence intervals (gray shadings) shows predictions of linear mixed models.

**Figure 2 insects-15-00266-f002:**
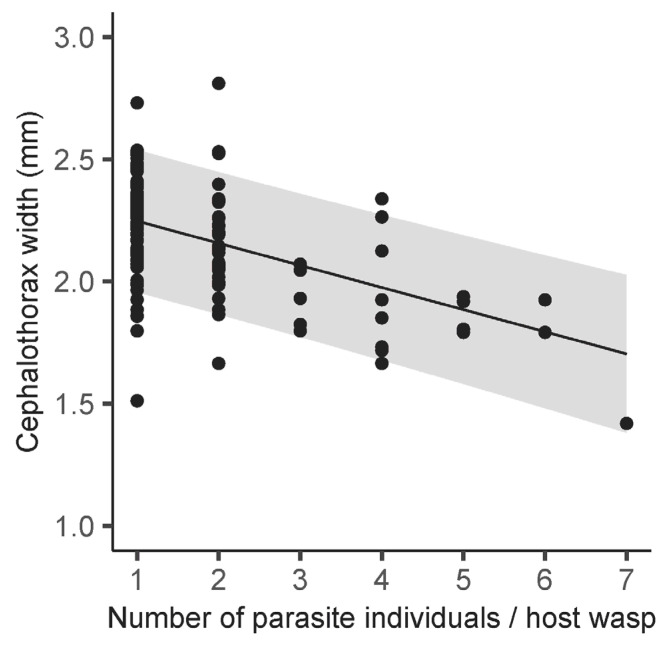
Relationship between number of female parasites per host wasp and cephalothorax width of female parasites. A line with 95% confidence intervals (gray shadings) shows predictions of linear mixed models.

**Figure 3 insects-15-00266-f003:**
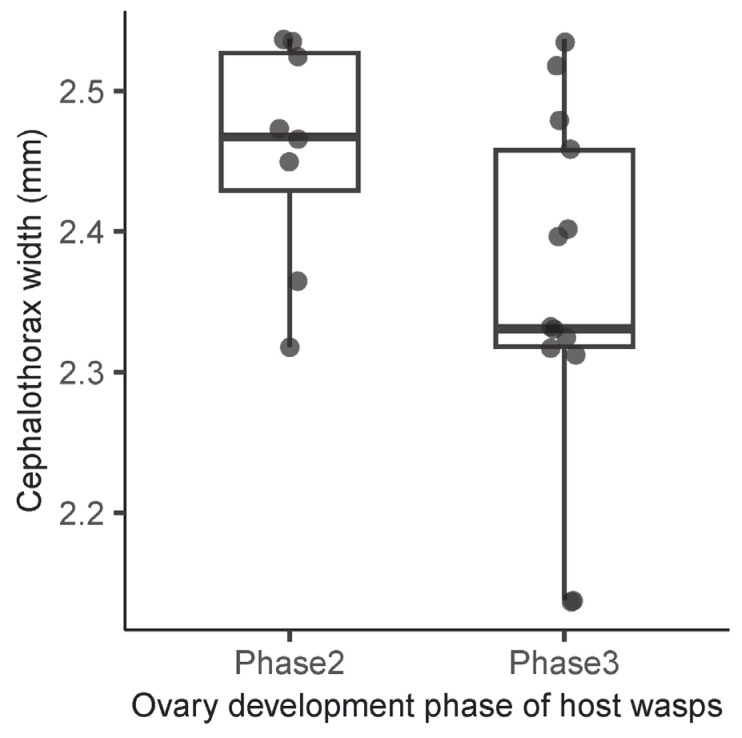
Relationship between cephalothorax width of *X. oxyodontes* and ovary development phases of host wasps. The horizontal lines denote median values, box boundaries represent interquartile ranges (25–75% percentiles), and whiskers indicate maximum and minimum values. Gray dots represent one parasite individual. See the text with regard to the phases.

**Table 1 insects-15-00266-t001:** Results of the most parsimonious linear mixed model for the cephalothorax width of female *Xenos* parasites.

Variable	Estimate	SE	t
(Intercept)	1.68	0.18	9.06
HW of host wasps	0.10	0.03	3.75
Number of parasites per host	−0.09	0.01	−6.87

## Data Availability

All data are provided in this paper.

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
