# Peer review of "Body Size of Female Strepsipteran Parasites (Strepsiptera, Xenidae, Xenos) Depends on Several Key Factors in a Vespine Wasp (Hymenoptera, Vespidae, Vespa)"

_insects, 2024, doi:10.3390/insects15040266_

Round 1

Reviewer 1 Report

Comments and Suggestions for Authors

Strepsiptera is internal parasite of other insects such as wasps and bees. In this study, the authors examined effects of season, host body size, multiparasitism and host reproductive status on the body size of Strepsipteran parasite females. The authors showed that the body size of Strepsipteran parasites is significantly influenced by host body size, number of parasites per host wasp, and host reproductive status, but not by season. Although this study is somewhat descriptive, the overall data presented here contribute to the advancement of our understanding of the biology of this "unique" endoparasite.

 The description and discussion of the results seem adequate. Overall the paper is an interesting, but I found this manuscript includes several minor points that should be considered by the authors.

(Minor point)

・Although the legends for Figures 1 and 2 mention gray shading in the figures, there are no such shadings in the graphs.

 ãƒ»Figure 3, the degree of host ovarian development is shown as phase 2 or phase 3, but it would be better to specifically explain how to differentiate between them. Are these Phase 2 and Phase 3 the same as Type 1 and Type 2 described in line 140?  

・Also, I couldn't find any gray dots in figure 3. 

・Line 282 Boll. Fac. Edu. à Bull. Fac. Edu. ??

・Line 293 Polybia paulista à Polybia paulista

・Please delete from page 9 onwards.

Author Response

Thank you for your kind comments and suggestions on our manuscript. I revised our manuscript according to your comments and suggestions. Please find the highlighted by red marks in the manuscript. 

Reviewer 2 Report

Comments and Suggestions for Authors

This study explores the development of Xenos oxyodontes. The author tried to test four variables of wasp host Vespa analis, and found that 3 of them have an impact on the development of X. oxyodontes. Generally speaking, no problems were found in the experimental data and statistics. The results also verified its hypothesis, which can be regarded as a paper of considerable quality.

The content of the article is clear and the experiment depth is sufficient. I am curious whether the parasitized wasps have preferences for the bait used in the experimental design, or healthy wasps have similiar reaction? I suggest that the journal editor accept this article in the future. I found only a few comments and if the editor accepts the article, the author should first check the following sections.

The following text requires further examination:

M&M

Line 119 (Makino & Yamashita, 1998) --> [12]

Line 132 Makino & Yamashita, 1998 --> Makino & Yamashita, 1998 [12]

Line 132 Kudo et al., 2024 --> Kudo et al., 2024 [17]

Results

Figure 1. (grey shadings) --> 1. No gray shadings are found in the figure, 2. Please indicate the scientific name

Figure 2. (grey shadings) --> 1. No gray shadings are found in the figure, 2. Please indicate the scientific name

Line 178 Xenos oxyodontes--> X. oxyodontes

Figure 3. gray dots --> No gray dots are found in the figure

References

Line 272 Xenos Strepsipterans--> Should be Xenos Strepsipterans?

Line 274 Xenos mouton -> Should be Xenos moutoni?

Line 293: Use italics for scientific names and journal abbreviations.

Line 307 Journal abbreviation in italics

Author Response

(The authors gave the same response as above.)
